# Uplifting Bandits

**Yu-Guan Hsieh**[*]
University of Grenoble Alpes
yu-guan.hsieh@univ-grenoble-alpes.fr

**Shiva Prasad Kasiviswanathan**
Amazon
kasivisw@gmail.com

**Branislav Kveton**
Amazon
bkveton@amazon.com

## Abstract

We introduce a multi-armed bandit model where the reward is a sum of multiple random variables, and each action only alters the distributions of some of them. After each action, the agent observes the realizations of all the variables. This model is motivated by marketing campaigns and recommender systems, where the variables represent outcomes on individual customers, such as clicks. We propose UCB-style algorithms that estimate the *uplifts* of the actions over a baseline. We study multiple variants of the problem, including when the baseline and affected variables are unknown, and prove sublinear regret bounds for all of these. We also provide lower bounds that justify the necessity of our modeling assumptions. Experiments on synthetic and real-world datasets show the benefit of methods that estimate the uplifts over policies that do not use this structure.

## 1 Introduction

Multi-armed bandit (MAB) is an important framework for sequential decision making under uncertainty [8, 19, 20]. In this problem, a learner repeatedly takes action and receives their rewards, while the outcomes of the other actions are unobserved. The goal of the learner is to maximize their cumulative reward over time by balancing exploration (select actions with uncertain reward estimates) and exploitation (select actions with high reward estimates). MAB has applications in online advertisement, recommender systems, portfolio management, and dynamic channel selection in wireless networks [7].

One prominent question in the MAB literature is how the dependencies between the actions can be exploited to improve statistical efficiency. Popular examples include *linear bandits* [11] and *combinatorial bandits* [9]. In this work, we study a different structured bandit problem with the following three features: (*i*) the reward is the sum of the payoffs of a fixed set of variables; (*ii*) these payoffs are observed; and (*iii*) each action only affects a small subset of the variables. This structure arises in many applications, as discussed below.

**(a) Online Marketing.** An ecommerce platform can opt among several marketing strategies (actions) to encourage customers to make more purchases on their website. As different customers can be sensitive to a different marketing strategy, regarding each of them as a variable, it is natural to expect that each action would likely affect only a subset of the variables. The payoff associated to a customer can be for example the revenue generated by that customer; then the reward is just the total revenue received by the platform.

**(b) Product Discount.** Consider a company that uses discount strategies to increase their sales. It is common to design discount strategies that only apply to a small subset of products. In this case, we

---

[*]Work done during an internship at Amazon.

36th Conference on Neural Information Processing Systems (NeurIPS 2022).

| Algorithm | UCB | UpUCB (b) | UpUCB | UpUCB-nAff | UpUCB-iLift |
|---|---|---|---|---|---|
| Affected variables known | No | Yes | Yes | No | No |
| Baseline payoffs known | No | Yes | No | No | No |
| Regret Bound | $\dfrac{Km^2}{\Delta}$ | $\dfrac{KL^2}{\Delta}$ | | $\dfrac{KL^2}{\Delta}$ | $\dfrac{K\,\mathrm{clip}(\Delta/\Delta_{\mathrm{up}}, L, m)^2}{\Delta}$ |

**Table 1:** Summary of our regret bounds for uplifting bandits. Constant and logarithmic factors are ignored throughout. For simplicity, we assume here all actions affect exactly $L$ of the $m$ variables and all the suboptimal actions have the same suboptimality gap $\Delta$. $K$ and $\Delta_{\mathrm{up}}$ are respectively the number of actions and a lower bound on individual uplift ($\Delta_{\mathrm{up}}$ is formally introduced in Appendix E). The operator clip restricts the value of its first variable to the range defined by its second and third variables, $\mathrm{clip}(x, \alpha, \beta) = \max(\alpha, \min(\beta, x))$.

can view the sales of each product as a variable and each discount strategy as an action, and assume that each action only has a significant impact on the sales of those products discounted by this action.

**(c) Movie Recommendation.** Consider a bandit model for movie recommendation where actions correspond to different recommendation algorithms, and the variables are all the movies in the catalog. For each user, define a set of binary payoffs that indicate whether the user watches a movie in the catalog, and the reward is the number of movies from the catalog that this user watches. Since a recommendation (action) for this user contains (promotes) only a subset of movies, it is reasonable to assume that only their associated payoffs are affected by that action.

**Our Contributions.** To begin with, we formalize an *uplifting bandit* model that captures the aforementioned structure, with the term *uplift* borrowed from the field of *uplift modeling* [15, 22] to indicate that the actions' effects are incremental over a certain baseline. The model is stochastic, that is, the payoffs of the variables follow an unknown distribution that depends on the chosen action. We study this problem under various assumptions on the learners' prior knowledge about **(1)** the baseline payoff of each variable and **(2)** the set of affected variables of each action. Our first result (Section 3) shows that when both **(1)** and **(2)** are known, a simple modification of the upper confidence bound (UCB) algorithm [3] (Algorithm UpUCB (b)) for estimating the uplifts has a much lower regret than its standard implementation. Roughly speaking, when $m$ is the number of variables and $L$ is the number of variables affected by each action, we get a $O(L^2)$ regret bound instead of $O(m^2)$. This results in a major reduction for $L \ll m$, and is a distinguishing feature of all our results.

Going one step further, we design algorithms that have minimax optimal regret bounds without assuming that either **(1)** or **(2)** is known. When **(1)** the baseline payoffs are unknown and **(2)** the affected variables are known, we compute differences of UCBs to estimate the uplifts (Algorithm UpUCB). In contrast to standard UCB methods, these differences are *not* optimistic, in the sense they are not high-probability upper bounds on the estimated quantity. This construction reflects the fact that the feedback for any single action also provides information about the baseline. When **(2)** the affected variables are also unknown, we identify them on the fly to maintain suitable estimates of the uplifts. We study two approaches, which differ in what they know about the affected variables. Algorithm UpUCB-nAff (Section 6) knows an upper bound on number of affected variables, whereas Algorithm UpUCB-iLift (Appendix E) knows a lower bound on individual uplift. Our regret bounds are summarized in Table 1. These results are further complemented with lower bounds that justify the need for our modeling assumptions (Section 4). To demonstrate the generality of our setup and how our algorithmic ideas extend beyond vanilla multi-armed bandits, we discuss contextual extensions of our model in Section 7. The experiments in Section 8 confirm the benefit of our approach.

**Related Work.** The goals of both uplifting modeling and MAB is to help selecting the optimal action. The former achieves it by modelling the incremental effect of an action on an individual's behavior. Despite this apparent connection between the two concepts, few papers explicitly link them together. We believe that this is because uplifting can be solved by classic bandit algorithms with a redefined reward. This approach was taken in [6, 23].

Instead, our paper focuses on bandit problems in which estimating the uplifts improves the statistical efficiency of the algorithms, and this is made possible thanks to the 'sparsity' of the actions' effects. Prior to our work, sparsity assumptions in bandits primarily concerned the sparsity of the parameter vectors in linear bandits [2, 5, 16]. A notable exception is Kwon et al. [18] who studied a variant of the MAB where the sparsity is reflected by the fact that the number of arms with positive reward is small. Our work is orthogonal to all of these in that we look at a different form of sparsity. As

we will see in Section 7, while sparsity in parameter can be a cause of sparsity in action effect, the improvement of regret is established with a different mechanism.

The additive structure of the reward and observability of individual payoffs also suggest some similarity between our model and that of combinatorial semi-bandits [10, 17, 21]. There, the learner selects at each round a subset of ground items and the reward is generally defined as the sum of the payoffs of the selected items. However, this seeming similarity comes with an important conceptual difference: the set of items, or variables, for which we observe the outcomes are selected in semi-bandits, while for us they are fixed and inherent to the reward generation mechanism. As an example, in Application (c), combinatorial bandit models optimize the number of movies that are watched among the recommended ones, while we also consider movies that are not recommended. In fact, as argued by Wang et al. [24], a recommendation is only effective if the user actually watches the movie and would not watch it without the recommendation. We refer the readers to Appendix B for more technical details and also a comparison with the causal bandit model.

**Organization.** We introduce our uplifting bandit model along with its various variations in Section 2. Over Sections 3, 5, 6, we provide regret bounds for these variations, with a lower bound presented in Section 4. We discuss contextual extensions in Section 7 and experimental results in Section 8.

## 2 Problem Description

We start by formally introducing our uplifting bandit model. We illustrate it in Fig. 1 and summarize our notation in Appendix A. Contextual extensions of this model are discussed in Section 7.

A $(K, m)$-uplifting bandit is a stochastic bandit with $K$ actions and $m$ underlying variables. Each action $a \in \mathcal{A} := \{1, ..., K\}$ is associated with a distribution $\mathcal{P}^a$ on $\mathbb{R}^m$. At each round $t$, the learner chooses an action $a_t \in \mathcal{A}$ and receives reward $r_t = \sum_{i \in \mathcal{V}} y_t(i)$ where $\mathcal{V} := \{1, ..., m\}$ is the set of all variables and $y_t = (y_t(i))_{i \in \mathcal{V}} \sim \mathcal{P}^{a_t}$ is the payoff vector.[2] Our model is distinguished by two assumptions that we describe below.

**(I) Limited Number of Affected Variables.** Let $\mathcal{V}^a \subseteq \mathcal{V}$ be the subset of variables affected by action $a$ and $\mathcal{P}^0$ be the baseline distribution that the variables' payoffs follow when no action is taken. By definition $\mathcal{P}^a$ and $\mathcal{P}^0$ have the same marginal distribution on $\overline{\mathcal{V}^a} := \mathcal{V} \setminus \mathcal{V}^a$, the variables unaffected by action $a$.[3] While the above condition is always satisfied with $\mathcal{V}^a = \mathcal{V}$, we are interested in the case of $L^a := \mathrm{card}(\mathcal{V}^a) \ll m$, meaning only a few variables are affected by $a$. We define $L$ as a uniform bound on $L^a$, so that $\max_{a \in \mathcal{A}} L^a \leq L$. For convenience of notation, we write $[T] := \{1, ..., T\}$ and $\mathcal{A}_0 = \mathcal{A} \cup \{0\}$, where $0$ is used for all the quantities related to the baseline.

**(II) Observability of Individual Payoff.** In addition to the reward $r_t$, we assume that the learner observes all the payoffs $(y_t(i))_{i \in \mathcal{V}}$ after an action is taken in round $t$.

**Uplift and Noise.** Let $y^a = (y^a(i))_{i \in \mathcal{V}}$ be a random variable with distribution $\mathcal{P}^a$. We define $\mu^a(i) = \mathbb{E}[y^a(i)]$ and $\xi^a(i) = y^a(i) - \mu^a(i)$ respectively as the expected value and the noise component of $y^a(i)$. We use $\mu^a$ (resp. $\mu^0$) to denote the vector of $(\mu^a(i))_{i \in \mathcal{V}}$ (resp. $(\mu^0(i))_{i \in \mathcal{V}}$), and refer to $\mu^0$ as the baseline payoffs. The *individual* uplift associated to a pair $(a, i) \in \mathcal{A} \times \mathcal{V}$ is defined as $\mu^a_{\mathrm{up}}(i) = \mu^a(i) - \mu^0(i)$. An individual uplift can be *positive* or *negative*. We obtain the (total) uplift of an action by summing its individual uplifts over all the variables affected by that action

$$r^a_{\mathrm{up}} = \sum_{i \in \mathcal{V}^a} \mu^a_{\mathrm{up}}(i) = \sum_{i \in \mathcal{V}^a} (\mu^a(i) - \mu^0(i)). \tag{1}$$

Let $r^a = \sum_{i \in \mathcal{V}} \mu^a(i)$ be the expected reward of an action or of pure observation. We also have $r^a_{\mathrm{up}} = r^a - r^0$ since $\mu^a(i) = \mu^0(i)$ as long as $i \notin \mathcal{V}^a$.

A real-value random variable $X$ is said to be $\sigma$-sub-Gaussian if for all $\gamma \in \mathbb{R}$, it holds $\mathbb{E}[\exp(\gamma X)] \leq \exp(\sigma^2 \gamma^2 / 2)$. Throughout the paper, we assume that $\xi^a(i)$ is 1-sub-Gaussian for all $a \in \mathcal{A}_0$ and $i \in \mathcal{V}$. Note that we do not assume that $(y^a(i))_{i \in \mathcal{V}}$ are independent, i.e., the elements in the noise vector $(\xi^a(i))_{i \in \mathcal{V}}$ may be correlated, for the following two reasons:

---

[2]The terms *reward* and *payoff* distinguish $r_t$ and $(y_t(i))_{i \in \mathcal{V}}$.

[3]If the actions in fact have small impact on the variables in $\overline{\mathcal{V}^a}$, our model is misspecified and incurs additional linear regret whose size is proportional to the impact of the actions on $\overline{\mathcal{V}^a}$. An interesting question is how we can design algorithms that self-adapt to the degree of misspecification.

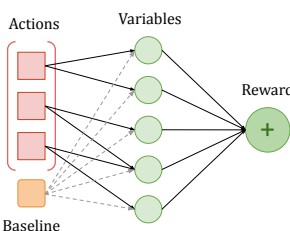

Actions, Variables, Reward, Baseline

**Figure 1:** Illustration of the uplifting bandit model. This example has $K = 3$ actions, $m = 5$ variables and each action affects $L^a = 2$ variables. Dash lines indicate the variables' payoffs follow the baseline distribution $\mathcal{P}^0$ by default.

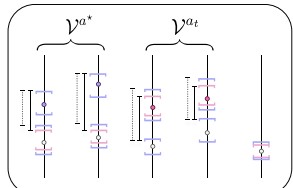

$\mathcal{V}^{a^\star}$ $\quad$ $\mathcal{V}^{a_t}$

(a) UPUCB: After suboptimal action $a_t$ is taken, the confidence intervals of $(\mu^a(i))_{i \in \mathcal{V}^{a_t}}$ and $(\mu^0(i))_{i \in \mathcal{V}^{a^\star} \setminus \mathcal{V}^{a_t}}$ shrink (from purple to pink). Hence the uplifting index of $a_t$ decreases while that of $a^\star$ increases (from dash to solid).

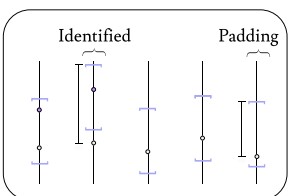

Identified $\quad$ Padding

(b) UPUCB-nAff (b): To compute the uplifting index of an action, we identify a set of variables whose associated confidence intervals do not contain the baseline payoff and pad it with the variables with the largest UCBs on uplifts.

**Figure 2:** Explanation about UPUCB and UPUCB-nAff (b) using model from Fig. 1. The five vertical bars correspond to the five actions while the $y$ axis corresponds to the value of the payoff.

- The independence assumption is not always realistic. In our first example, it excludes any potential correlation between two customers' purchases.

- While a learner can exploit their knowledge on the noise covariance matrix to reduce the regret, as for example shown by Degenne and Perchet [12], incorporating such knowledge complicates the algorithm design and analysis. However, we believe this is an interesting future direction to pursue.

**Regret.** The learner's performance is characterized by their regret. To define it, we denote by $a^\star \in \arg\max_{a \in \mathcal{A}} r^a$ an action with the largest expected reward and by $r^\star = r^{a^\star} = \max_{a \in \mathcal{A}} r^a$ the largest expected reward. The regret of the learner after $T$ rounds is then given by

$$\text{Reg}_T = r^\star T - \sum_{t=1}^{T} r^{a_t} = \sum_{a \in \mathcal{A}} \sum_{t=1}^{T} \mathbb{1}\{a_t = a\}\Delta^a, \tag{2}$$

where $\Delta^a = r^\star - r^a$ is the suboptimality gap of action $a$. In words, we compare the learner's cumulative (expected) reward against the best we can achieve by taking an optimal action at each round. In the following, we also write $\Delta = \min_{a \in \mathcal{A}, \Delta^a > 0} \Delta^a$ for the minimum non-zero suboptimality gap and refer to it as the suboptimality gap of the problem.

**UCB for Uplifting Bandits.** The UCB algorithm [4], at each round, constructs an *upper confidence bound* (UCB) on the expected reward of each action and chooses the action with the highest UCB. When applied to our model, we get a regret of $\mathcal{O}(Km^2 \log T / \Delta)$, as the noise in the reward is $m$-sub-Gaussian (recall that we do not assume independence of the payoffs). However, this approach completely ignores the structure of our problem. As we show in Section 3, we can achieve much smaller regret by focusing on the uplifts.

**Problem Variations.** In the following sections, we study several variants of the above basic problem that differ in the prior knowledge about $(\mathcal{P}^a)_{a \in \mathcal{A}_0}$ that the learner possesses.

**(a) Knowledge of Baseline Payoffs.** We consider two scenarios based on whether the learner knows the baseline payoffs $\mu^0 := (\mu^0(i))_{i \in \mathcal{V}}$ or not.

**(b) Knowledge of Affected Variables.** We again consider two scenarios based on whether the learner knows the affected variables associated with each action $(\mathcal{V}^a)_{a \in \mathcal{A}}$ or not.

## 3 Case of Known Baseline and Known Affected Variables

We start with the simplest setting where both the affected variables and the baseline payoffs are known. To address this problem, we make the crucial observation that for any two actions, the difference in their rewards equals to that of their uplifts. As an important consequence, uplift maximization has the same optimal action as total reward maximization, and we can replace rewards by uplifts in the definition of the regret (2). Formally, we write $r^\star_{\text{up}} = r^{a^\star}_{\text{up}} = \max_{a \in \mathcal{A}} r^a_{\text{up}}$ for the largest uplift; then $\text{Reg}_T = r^\star_{\text{up}} T - \sum_{t=1}^{T} r^{a_t}_{\text{up}}$. By making this transformation, we gain in statistical efficiency because

$r_{\text{up}}^a = \mathbb{E}[\sum_{i \in \mathcal{V}^a}(y^a(i) - \mu^0(i))]$ can now be estimated much more efficiently under our notion of sparsity. Since both $\mu^0$ and $(\mathcal{V}^a)_{a \in \mathcal{A}}$ are known, we can directly construct a UCB on $r_{\text{up}}^a$. For this, we define for all rounds $t \in [T]$, actions $a \in \mathcal{A}$, and variables $i \in \mathcal{V}$ the quantities

$$N_t^a = \sum_{s=1}^t \mathbb{1}\{a_s = a\}, \quad c_t^a = \sqrt{\frac{2\log(1/\delta')}{N_t^a}}, \quad \hat{\mu}_t^a(i) = \sum_{s=1}^t \frac{y_s(i)\mathbb{1}\{a_s = a\}}{\max(1, N_t^a)}, \quad (3)$$

where $\delta' > 0$ is a tunable parameter. In words, $N_t^a$, $\hat{\mu}_t^a(i)$, and $c_t^a$ represent respectively the number of times that action $a$ is taken, the empirical estimate of $\mu^a(i)$, and the associated radius of confidence interval calculated at the end of round $t$. The UCB for action $a \in \mathcal{A}$ and variable $i \in \mathcal{V}^a$ at round $t$ is $U_t^a(i) = \hat{\mu}_{t-1}^a(i) + c_{t-1}^a$. We further define $\tau_t^a = \sum_{i \in \mathcal{V}^a}(U_t^a(i) - \mu^0(i))$ as the *uplifting index*, and refer to UPUCB (b) as the algorithm that takes an action with the largest uplifting index $\tau_t^a$ at each round $t$ (Algorithm 3, Appendix A). Here, the suffix (b) indicates that the method operates with the knowledge of the baseline payoffs.

Since UPUCB (b) is nothing but a standard UCB with transformed rewards $r_t' = \sum_{i \in \mathcal{V}^{a_t}}(y_t(i) - \mu^0(i))$, and $\mathbb{E}[r_t'] = r_{\text{up}}^{a_t}$ defines exactly the same regret as the original reward, a standard analysis for UCB yields the following result.

**Proposition 1.** *Let $\delta' = \delta/(2KT)$. Then the regret of* UPUCB *(b) (Algorithm 3, Appendix A), with probability at least $1 - \delta$, satisfies*

$$\text{Reg}_T \leq \sum_{a \in \mathcal{A}: \Delta^a > 0} \left( \frac{8(L^a)^2 \log(2KT/\delta)}{\Delta^a} + \Delta^a \right). \quad (4)$$

As expected, the regret bound does not depend on $m$ and scales with $L^2$. This is because the transformed reward of action $a$ is only $L^a$-sub-Gaussian. The improvement is significant when $L \ll m$. However, this also comes at a price, as both the baseline payoffs and affected variables need to be known. We address these shortcomings in Sections 5 and 6. On a side note, we remark that observing the aggregated payoff $\sum_{i \in \mathcal{V}^{a_t}} y_t(i)$ is sufficient for UPUCB (b), but this will not be the case for the other algorithms presented in our paper.

**Estimating Baseline Payoffs from Observational Data.** In practice, the baseline payoffs $\mu^0$ can be estimated from observational data, which gives confidence intervals that $\mu^0(i)$ lie in with high probability. The uplifting indices can then be constructed by subtracting the lower confidence bounds of $\mu^0(i)$ from $U_t^a(i)$. If the number of samples is $n$, the widths of the confidence intervals are in $\mathcal{O}(1/\sqrt{n})$. Identification of $a^*$ is possible only when the sum of these widths over $L$ variables is at most $\Delta$, which requires $n = \Omega(L^2/\Delta^2)$. Otherwise, these errors persist at each iteration and $n$ must be in the order of $T$ to ensure $\mathcal{O}(\sqrt{T})$ regret.

**Gap-Free Bounds.** In Proposition 1, we state a high-probability instance-dependent regret bound, and we will continue to do so for all the regret upper bounds that we present for the non-contextual variants. This type of result can be directly transformed to bound on $\mathbb{E}[\text{Reg}_T]$ by taking $\delta = 1/T$. Following the routine of separating $\Delta^a$ into two groups depending on their scale, most of our proofs can be modified to obtain a gap-free bound, which is usually in the order of $\mathcal{O}(L\sqrt{KT\log(T)})$. We will not state these results to avoid unnecessary repetitions.

## 4 Lower Bounds

In this section, we shortly discuss the necessity of our modeling assumptions for obtaining the improved regret bounds of Proposition 1. The complete discussion appears in Appendix D.

Intuitively, the regret can be improved both because the noise in the effect of an action is small, and because the observation of this effect does not heavily deteriorate with noise. These two points correspond respectively to assumptions (I) and (II). Moreover, the knowledge on $(\mathcal{V}^a)_{a \in \mathcal{A}}$ allows the learner to distinguish between problems with different structures. Without such distinction, there is no chance that the learner can leverage the underlying structure. Therefore, the aforementioned three points are crucial for obtaining (4). Below, we establish this formally for algorithms that are *consistent* [19] over the class of 1-sub-Gaussian uplifting bandits, which means the induced regret of the algorithm on any uplifting bandit with 1-sub-Gaussian noise satisfies $\text{Reg}_t = o(t^p)$ for all $p > 0$.

**Proposition 2.** *Let $\pi$ be a consistent algorithm over the class of 1-sub-Gaussian uplifting bandits that at most uses the knowledge of $\mathcal{P}^0$, $(\mathcal{V}^a)_{a \in \mathcal{A}}$, and the fact that the noise is 1-sub-Gaussian. Let $K, m > 0$ and sequence $(L^a, \Delta^a)_{1 \leq a \leq K} \in ([m] \times \mathbb{R}_+)^K$ satisfy $\Delta^1 = 0$. Assume either of the following holds.*

*(a) $L^a = m$ for all $a \in \mathcal{A}$, so that in the bandits considered below all actions affect all variables.*

*(b) Only the reward is observed.*

*(c) The algorithm $\pi$ does not make use of any prior knowledge about the arms' expected payoffs $(\mu^a)_{a \in \mathcal{A}}$ (in particular, the knowledge of $(\mathcal{V}^a)_{i \in \mathcal{V}}$ is not used by $\pi$).*

*Then, there exists a 1-sub-Gaussian $(K, m)$-uplifting bandit whose suboptimality gaps and numbers of affected variables are respectively $(\Delta^a)_{a \in \mathcal{A}}$ and $(L^a)_{a \in \mathcal{A}}$, such that the regret induced by $\pi$ on it satisfies:* $\liminf_{T \to +\infty} \frac{\mathbb{E}[\mathrm{Reg}_T]}{\log T} \geq \sum_{a \in \mathcal{A}: \Delta^a > 0} \frac{2m^2}{\Delta^a}$.

Proposition 2 states an instance-dependent lower-bound for a learner that may be equipped with full knowledge of the baseline distribution.[4] Its proof is presented in Appendix D. At this point, what remains unclear is whether similar improvement of the regret is still possible when the baseline is unknown or when the learner only has access to more restricted knowledge than $(\mathcal{V}^a)_{a \in \mathcal{A}}$. We give affirmative answers to these two questions in the next two sections.

## 5    Case of Unknown Baseline

In this section, we consider the situation where the learner knows the sets of affected variables $(\mathcal{V}^a)_{a \in \mathcal{A}}$, but not the baseline payoffs. Since the actual uplift at each round is never observed, the uplifts of the actions can hardly be estimated directly in this case. Instead, we follow a two-model approach. Leveraging the fact that $\mathcal{P}^a$ and $\mathcal{P}^0$ have the same marginal distribution on $\overline{\mathcal{V}^a}$, we can estimate the baseline payoffs by aggregating information from the feedback of different actions. This leads to

$$N_t^0(i) = \sum_{s=1}^{t} \mathbb{1}\{i \notin \mathcal{V}^{a_s}\}, \quad c_t^0(i) = \sqrt{\frac{2 \log(1/\delta')}{N_t^0(i)}}, \quad \hat{\mu}_t^0(i) = \frac{\sum_{s=1}^{t} y_s(i) \, \mathbb{1}\{i \notin \mathcal{V}^{a_s}\}}{\max(1, N_t^0(i))}. \quad (5)$$

Compared to (3), we notice that both $N_t^0$ and $c_t^0$ are functions of $i$. This is because for each taken action $a$, we only increase the counters $N_t^0(i)$ for those $i \notin \mathcal{V}^a$, which causes a non-uniform increase of $(N_t^0(i))_{i \in \mathcal{V}}$. Then, by looking at all the rounds that variable $i$ is not influenced by the chosen action, we manage to compute $\hat{\mu}_t^0(i)$, an estimate of $\mu^0(i)$. To proceed, we define the following UCB indices for all the pairs $(a, i) \in \mathcal{A}_0 \times \mathcal{V}$

$$U_t^a(i) = \begin{cases} 0 & \text{if } a = 0 \text{ and } i \in \bigcap_{a \in \mathcal{A}} \mathcal{V}^a, \\ \hat{\mu}_{t-1}^a(i) + c_{t-1}^a & \text{if } a \in \mathcal{A} \text{ and } i \in \mathcal{V}^a, \\ \hat{\mu}_{t-1}^0(i) + c_{t-1}^0(i) & \text{otherwise.} \end{cases} \quad (6)$$

The second and the third lines of (6) contain the usual definition of UCBs using the empirical estimates and the radii of the confidence intervals defined in (3) and (5). In the special case that a variable is affected by all the actions, it is impossible to estimate $U^0(i)$ but it is enough to compare $U^a(i)$ directly against $U^{a'}(i)$ for any two actions $a, a' \in \mathcal{A}$, so we just set $U_t^0(i)$ to 0 in this case.

We outline the proposed method, UPUCB, in Algorithm 1. The uplifting indices are given by $\tau_t^a = \sum_{i \in \mathcal{V}^a}(U_t^a(i) - U_t^0(i))$. It may be counter-intuitive to compare the differences between two UCBs. Indeed, $\tau_t^a$ is no longer an optimistic estimate of the uplifting effect $r_{\mathrm{up}}^a$, but it captures the essential trade-off between learning action $a$'s payoffs and learning the baseline $\mu^0$. To provide some intuition, we give an informal justification of UPUCB in Fig. 2a: If a suboptimal action $a$ is taken in round $t$, the estimates of all $U_t^a(i)$ move closer to the actual mean from above. As a result, $\tau_t^a$ decreases, since all $U_t^a(i)$ for $i \in \mathcal{V}^a$ do. Thus action $a$ is less likely to be taken next. Moreover, $\tau_t^{a^\star}$ increases, since $U_t^0(i)$ decrease for any $i$ affected by $a^\star$ but not $a$. Thus $a^\star$ is more likely to be taken next. The effectiveness of UPUCB is confirmed by the following theorem.

---

[4]Of course, the problem only becomes more challenging if the learner does not know the baseline distribution.

---
**Algorithm 1** UPUCB
---
1: **Input:** Error probability $\delta'$, the sets of variables each action affects $\{\mathcal{V}^a : a \in \mathcal{A}\}$
2: **Initialization:** Take each action once
3: **for** $t = K + 1, \ldots, T$ **do**
4:     Compute the UCB indices following (6)
5:     For $a \in \mathcal{A}$, set $\tau_t^a \leftarrow \sum_{i \in \mathcal{V}^a} (U_t^a(i) - U_t^0(i))$
6:     Select action $a_t \in \arg\max_{a \in \mathcal{A}} \tau_t^a$
---

**Theorem 1.** *Let $\delta' = \delta/(4KLT)$. Then the regret of* UPUCB *(Algorithm 1) with probability at least* $1 - \delta$, *satisfies:*

$$\mathrm{Reg}_T \leq \sum_{a \in \mathcal{A} : \Delta^a > 0} \left( \frac{8(L^a + L^{a^\star})^2 \log(4KLT/\delta)}{\Delta^a} + \Delta^a \right). \tag{7}$$

*Idea of proof.* The full proof of Theorem 1 is presented in Appendix C.2, and proceeds as following.

1. With concentration of measure, we show that with probability $1 - \delta$, it holds $|\hat{\mu}_t^a(i) - \mu^a(i)| \leq c_t^a$ and $|\hat{\mu}_t^0(i) - \mu^0(i)| \leq c_t^0(i)$ for all relevant estimates. It is thus sufficient to show that (7) holds when these inequalities are satisfied.

2. A suboptimal action $a$ can only be taken if its uplifting index is larger than that of $a^\star$, i.e., if $\sum_{i \in \mathcal{V}^a}(U_t^a(i) - U_t^0(i)) \geq \sum_{i \in \mathcal{V}^{a^\star}}(U_t^{a^\star}(i) - U_t^0(i))$. Rearranging, we get

$$\sum_{i \in \mathcal{V}^a} U_t^a(i) + \sum_{i \in \mathcal{V}^{a^\star} \setminus \mathcal{V}^a} U_t^0(i) \geq \sum_{i \in \mathcal{V}^{a^\star}} U_t^{a^\star}(i) + \sum_{i \in \mathcal{V}^a \setminus \mathcal{V}^{a^\star}} U_t^0(i). \tag{8}$$

Using the inequalities mentioned in the previous point and $\Delta^a = r_{\mathrm{up}}^{a^\star} - r_{\mathrm{up}}^a$, bounding the two sides of (8), we deduce $\Delta^a \leq 2L^a c_{t-1}^a + \sum_{i \in \mathcal{V}^{a^\star} \setminus \mathcal{V}^a} 2c_{t-1}^0(i)$.

3. Note that for all $i \in \mathcal{V}^{a^\star} \setminus \mathcal{V}^a$, whenever action $a$ is taken, the count of $N_s^0(i)$ also increases by 1. We have thus $N_{t-1}^0(i) \geq N_{t-1}^a$ and accordingly $c_{t-1}^0(i) \leq c_{t-1}^a$. We then get $\Delta^a \leq 2(L^a + L^{a^\star})c_{t-1}^a$. This allows us to bound the number of times that action $a$ is taken and conclude. $\square$

As in Proposition 1, the regret of Theorem 1 is in $\mathcal{O}(KL^2 \log T/\Delta)$. In fact, all decisions in UPUCB are made on uncertain estimates of at most $L$ variables; thus the statistical efficiency scales with $L$ and not $m$. A detailed comparison with Proposition 1 reveals that the difference is in the order of $K(L^{a^\star})^2 \log T/\Delta$; this is only significant when the optimal actions affect many more variables then the suboptimal ones. Hence, the price of not knowing the baseline payoffs is generally quite small.

## 6 Case of Unknown Affected Variables

Now we study the more challenging setting where the affected variables $(\mathcal{V}^a)_{a \in \mathcal{A}}$ are unknown to the learner. Proposition 2 states that improvement is impossible if the learner does not have any prior knowledge to exploit the structure. To circumvent this negative result, we study two weak assumptions motivated by practice: learner has access to (*i*) an upper bound on the number of affected variables or (*ii*) a lower bound on individual uplift. Due to space constraints, we only present the first setting here and defer the discussion of the other to Appendix E.

For the rest of this section, we assume that we know an upper bound on the number of affected variables $L$ (i.e., $L \geq \max_{a \in \mathcal{A}} L^a$). We design algorithms with $\mathcal{O}(KL^2 \log T/\Delta)$ regret bounds that takes $L$ as input. We consider the cases of known and unknown baseline payoffs.

**Known Baseline Payoffs.** To illustrate our ideas, we start by assuming that the baseline payoffs are known. We propose an optimistic algorithm that maintains a UCB on the total uplift with an overestimate in the order of $L$. Let $N_t^a$, $c_t^a$, and $\hat{\mu}_t^a(i)$ be defined as in (3). The UCB, uplifting indices, and the confidence intervals for each (action, variable) pair $(a, i) \in \mathcal{A} \times \mathcal{V}$ are

$$U_t^a(i) = \hat{\mu}_{t-1}^a(i) + c_{t-1}^a, \quad \rho_t^a(i) = U_t^a(i) - \mu^0(i), \quad \mathcal{C}_t^a(i) = [\hat{\mu}_{t-1}^a(i) - c_{t-1}^a, \hat{\mu}_{t-1}^a(i) + c_{t-1}^a]. \tag{9}$$

In the rest of the paper, we refer to $\rho_t^a(i)$ as the *individual uplifting index* of the pair $(a, i)$. It is

---

**Algorithm 2** UPUCB-nAff

---

1: **Input:** Error probability $\delta'$, Upper bound $L$ on the number of variables that each action affects
2: **Initialization:** Take each action once
3: **for** $t = K + 1, \ldots, T$ **do**
4:     Choose $b_t \in \arg\max_{a \in \mathcal{A}} N^a_{t-1}$ and compute UCBs and confidence intervals using (9)
5:     **for** $a \in \mathcal{A}$ **do**
6:         Set $\widehat{\mathcal{V}}^a_t \leftarrow \{i \in \mathcal{V} : \mathcal{C}^a_t(i) \cap \mathcal{C}^{b_t}_t(i) = \emptyset\}$
7:         For $i \in \mathcal{V}$, compute $\rho^a_t(i) \leftarrow U^a_t(i) - U^{b_t}_t(i)$
8:         Set $L^a_t \leftarrow \max(0, 2L - \mathrm{card}(\widehat{\mathcal{V}}^a_t))$ and $\mathcal{L}^a_t \leftarrow \arg\max_{\mathcal{L} \subseteq \mathcal{V} \setminus \widehat{\mathcal{V}}^a_t, \ \mathrm{card}(\mathcal{L}) \leq L^a_t} \sum_{i \in \mathcal{L}} \rho^a_t(i)$
9:         Compute uplifting index $\tau^a_t \leftarrow \sum_{i \in \widehat{\mathcal{V}}^a_t \cup \mathcal{L}^a_t} \rho^a_t(i)$
10:    Select action $a_t \in \arg\max_{a \in \mathcal{A}} \tau^a_t$

---

an overestimate of the individual uplift $\mu^a_{\mathrm{up}}(i)$. As for $\mathcal{C}^a_t(i)$, it is the confidence interval that $\mu^a(i)$ lies in with high probability. Our algorithm, UPUCB-nAff (b) with nAff for number of affected, leverages two important procedures to compute an optimistic estimate of the uplift $r^a_{\mathrm{up}}$: identification of affected variables, and padding with variables with the highest individual uplifting indices.

To begin, UPUCB-nAff (b) constructs the set of *identified variables* $\widehat{\mathcal{V}}^a_t = \{i \in \mathcal{V} : \mu^0(i) \notin \mathcal{C}^a_t(i)\}$ which is contained in $\mathcal{V}^a$ with high probability. In fact, by concentration of measure, with high probability $\mu^a(i) \in \mathcal{C}^a_t(i)$, in which case $\mu^0(i) \notin \mathcal{C}^a_t(i)$ indicates $i \in \mathcal{V}^a$. However, $\widehat{\mathcal{V}}^a_t$ is not guaranteed to capture all the affected variables, so we also need to provide an upper bound for $\sum_{i \in \mathcal{V}^a \setminus \widehat{\mathcal{V}}^a_t} \mu^a_{\mathrm{up}}(i)$, the uplift contributed by the unidentified affected variables. Since the individual uplifting index $\rho^a_t(i)$ is in fact a UCB on the individual uplift $\mu^a_{\mathrm{up}}(i)$ here and $\mathrm{card}(\mathcal{V}^a) \leq L$, we can simply choose the $L - \mathrm{card}(\widehat{\mathcal{V}}^a_t)$ variables in $\mathcal{V} \setminus \widehat{\mathcal{V}}^a_t$ with the largest $\rho^a_t(i)$. Let us refer to this set as $\mathcal{L}^a_t$. We then get a proper UCB on the uplift of action $a$ by computing $\tau^a_t = \sum_{i \in \widehat{\mathcal{V}}^a_t \cup \mathcal{L}^a_t} \rho^a_t(i)$. This process is summarized in Algorithm 4 in Appendix A and illustrated in Fig. 2b.

**Unknown Baseline Payoffs.** Now we focus on the most challenging setting, where also the baseline payoffs are unknown. In this case, neither the sets of identified variables nor the uplifting indices of UPUCB-nAff (b) can be defined. We also cannot estimate the baseline payoffs using (5) since the sets of affected variables are unknown. To overcome these challenges, we note that for any two actions $a, a' \in \mathcal{A}$, $\mu^a$ and $\mu^{a'}$ only differ on $\mathcal{V}^a \cup \mathcal{V}^{a'}$, and $\mathrm{card}(\mathcal{V}^a \cup \mathcal{V}^{a'}) \leq 2L$. Therefore, $\mu^a$ and $\mu^{a'}$ differ in at most $2L$ variables, and we recover a similar problem structure by taking the payoffs of any action as the baseline.

Combining this idea with the elements that we have introduced previously, we obtain UPUCB-nAff (Algorithm 2). In each round, UPUCB-nAff starts by picking a most frequently taken action $b_t$ (Line 4) whose payoffs are treated as the baseline in that round. Then, in Line 6, UPUCB-nAff chooses variables that are guaranteed to be either in $\mathcal{V}^a$ or $\mathcal{V}^{b_t}$. This generalizes the identification step of UPUCB-nAff (b). The individual uplifting indices $\rho^a_t(i) = U^a_t(i) - U^{b_t}_t(i)$ are computed in Line 7. The differences of UCBs are inspired by a similar construction in UPUCB. Line 8 constitutes the padding step, during which variables with the highest uplifting indices are selected, and finally in Line 9 we combine the above to get the uplifting index of the action. To see why UPUCB-nAff is similar to UPUCB-nAff (b), suppose that one action has been taken frequently. Then the baseline payoffs are precisely estimated and do not change much between consecutive rounds.

**Regret.** Both UPUCB-nAff (b) and UPUCB-nAff choose $\mathcal{O}(L)$ variables for estimating the uplift of an action, and the decisions are based on these estimates. Therefore, the statistical efficiencies of these algorithms only scale with $L$ and not $m$. This in turn translates into an improvement of the regret, as demonstrated by the theorem below.

**Theorem 2.** *Let $\delta' = \delta/(2KmT)$. Then the regret of* UPUCB-nAff *(Algorithm 2) (resp.* UPUCB-nAff *(b), Algorithm 4), with probability at least $1 - \delta$, satisfies:*

$$\mathrm{Reg}_T \leq \sum_{a \in \mathcal{A}: \Delta^a > 0} \left( \frac{\alpha L^2 \log(2KmT/\delta)}{\Delta^a} + \Delta^a \right), \tag{10}$$

*where $\alpha = 512$ (resp. 32) in the above inequality.*

*Idea of proof.* The full proof of Theorem 2 is presented in Appendix C.3. We outline below the main steps to prove the result for UPUCB-nAff (b).

1. With concentration of measure, with probability $1 - \delta$ it holds $\mu^a(i) \in \mathcal{C}_t^a(i)$. Then, as explained in the text, $\widehat{\mathcal{V}}_t^a \subseteq \mathcal{V}^a$ and $\tau_t^a$ is an upper bound on $r_{\text{up}}^a$; especially $\tau_t^{a^\star} \geq r_{\text{up}}^{a^\star}$.

2. For $i \notin \widehat{\mathcal{V}}_t^a$, by definition $\mu^0(i) \in \mathcal{C}_t^a$, from which we deduce $0 \leq \rho_t^a(i) \leq 2c_{t-1}^a$. With $\widehat{\mathcal{V}}_t^a \subseteq \mathcal{V}^a$ and $\mathcal{L}_t^a \subseteq \mathcal{V} \setminus \widehat{\mathcal{V}}_t^a$ we can then write

$$\tau_t^a = \sum_{i \in \widehat{\mathcal{V}}_t^a \cup \mathcal{L}_t^a} \rho_t^a(i) \leq \sum_{i \in \mathcal{V}^a} \rho_t^a(i) + \sum_{i \in \mathcal{L}_t^a} \rho_t^a(i) \leq r_{\text{up}}^a + 4Lc_{t-1}^a.$$

We have also used $\text{card}(\mathcal{V}^a) \leq L$, $\text{card}(\mathcal{L}_t^a) \leq L$, and $\rho_t^a(i) \leq \mu^a(i) + 2c_{t-1}^a$.

3. Whenever a suboptimal action $a$ is taken, we have $\tau_t^a \geq \tau_t^{a^\star}$. Combined with the two previous point we then deduced $\Delta^a \leq 4Lc_{t-1}^a$. Subsequently, we bound the number of times that each suboptimal action $a$ is taken to conclude. $\square$

The proof of Theorem 2 is notable for two reasons. First, tracking of the identified variables guarantees that the uplifting index $\tau_t^a$ does not overestimate the uplift $r_{\text{up}}^a$ too much. Take UPUCB-nAff (b) as an example. An alternative to constructing a UCB on $r_{\text{up}}^a$ is to choose the $L$ variables with the highest individual uplifting indices $\rho_t^a(i)$. However, this would result in a severe overestimate when a negative individual uplift is present. Second, to prove (10) for UPUCB-nAff, we use that the widths of confidence intervals of the chosen $b_t$ are always smaller than those of the taken action. This is ensured by taking $b_t$ as the most frequent action (Line 4 in Algorithm 2).

## 7    Contextual Extensions

In this section, we briefly discuss potential contextual extensions of our model; a detailed case study is presented in Appendix F. As in contextual bandits, context is a side information that helps the learner to make a more informed decision, which results in a higher reward. To incorporate context, one possibility is to associate each variable with a feature vector $x_t(i) \in \mathbb{R}^d$. The subscript $t$ indicates that the context can change from one round to another. We also associate each action with a function $f^a$ so that the expected payoff of action $a$ acting on a variable with feature $x_t(i)$ is $f^a(x_t(i))$. The expected reward of choosing $a$ at round $t$ is then $r^a(x_t) = \sum_{i \in \mathcal{V}} f^a(x_t(i))$. The optimal action in round $t$ is $a_t^\star = \arg\max_{a \in \mathcal{A}} r^a(x_t)$ and the regret of a learner that takes the actions $(a_t)_{t \in [T]}$ is given by $\text{Reg}_T = \sum_{t=1}^T \sum_{i \in \mathcal{V}} (f^{a_t^\star}(x_t(i)) - f^{a_t}(x_t(i)))$.

The key structure in our model (Section 2) is that there exists a baseline payoff vector $\mu^0$ such that for any given action $a$, $\mu^a(i) = \mu^0(i)$ holds for most $i \in \mathcal{V}$. Given context, this translates into the existence of a baseline function $f^0$ such that for any given $a$ and $t$, $f^a(x_t(i)) = f^0(x_t(i))$ holds for most $i \in \mathcal{V}$. The uplift of action $a$ is defined as $r_{\text{up}}^a(x_t) = \sum_{i \in \mathcal{V}} f^a(x_t(i)) - f^0(x_t(i))$.

For concreteness, let us consider a model with linear payoffs. Then, each action is associated with an unknown parameter $\theta^a$ and the expected payoff is the scalar product of $\theta^a$ and the feature of the variable, i.e., $f^a(x_t(i)) = \langle \theta^a, x_t(i) \rangle$. We also assume the aforementioned equality to hold for the baseline function $f^0$ and we use $\mathcal{V}_t^a = \{i \in \mathcal{V} : \langle \theta^a, x_t(i) \rangle \neq \langle \theta^0, x_t(i) \rangle\}$ for the variables affected by action $a$ at round $t$. One sufficient condition for $\mathcal{V}_t^a$ to be small is sparsity in both the parameter difference $\theta_{\text{up}}^a = \theta^a - \theta^0$ and the context vector $x_t(i)$. In fact, $\langle \theta^a, x_t(i) \rangle = \langle \theta^0, x_t(i) \rangle$ as long as the supports of $\theta_{\text{up}}^a$ and $x_t(i)$ are disjoint. We assume $\text{card}(\mathcal{V}_t^a)$ is uniformly bounded by $L$ below.

Clearly, our algorithms can be directly applied as long as we can construct a UCB on $\langle \theta^a, x_t(i) \rangle$. This can for example be done using the construction of linear UCB [1]. In this way, the decision of the learner is again based on the uncertain estimates of at most $\mathcal{O}(L)$ variables, and we expect similar improvements as in our earlier theorems. As an example, when both $\theta^0$ and $\mathcal{V}_t^a$ are known, UPUCB (b) adapted to this situation constructs UCB for $\sum_{i \in \mathcal{V}_t^a} \langle \theta_{\text{up}}^a, x_t(i) \rangle$, and it is straightforward to show that the regret of such algorithm can be as small as $\mathcal{O}(Ld\sqrt{KT})$. In contrast, if the learner works directly with the total reward, the regret is in $\mathcal{O}(md\sqrt{KT})$.[5]

---

[5] We present gap-free bounds here and thus we get $L$ versus $m$ in the place of $L^2$ versus $m^2$. As in previous sections, these bounds apply to a potentially dependent noise.

# 8 Numerical Experiments

In this section, we present numerical experiments to demonstrate the benefit of estimating uplifts in our model. We compare our methods introduced in Sections 3, 5 and 6 against UCB and Thompson sampling with Gaussian prior and Gaussian noise that only use the observed rewards $(r_t)_{t \in [T]}$. To ensure a fair comparison, we tune all the considered algorithms and report results for the parameters that yield the best average performance. The detailed procedure, and additional experimental details and results, are provided in Appendices G and H. The experiments for the contextual extension in Section 7 are presented in Appendix F.3.

**Gaussian Uplifting Bandit.** We first study our algorithms in a Gaussian uplifting bandit with $K = 10$ actions, $m = 100$ variables, and $L^a = 10$ for all $a \in \mathcal{A}$ meaning that each action affects 10 variables.[6] The expected payoffs are contained in $[0, 1]$, and the covariance matrix of the noise is taken the same for all the actions. The suboptimality gap of the created problem is around 0.2, and the variance of the total noise $\sum_{i \in \mathcal{V}} \xi^a(i)$ is around 80.

**Bernoulli Uplifting Bandit with Criteo Uplift.** We use the Criteo Uplift Prediction Dataset [13] with 'visit' as the outcome variable to build a Bernoulli uplifting bandit, where the payoff of each variable has a Bernoulli distribu-

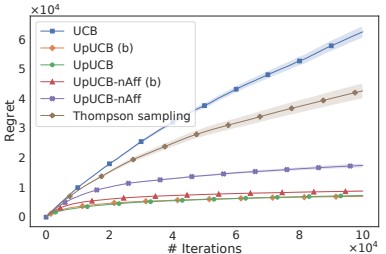

(a) Gaussian uplifting

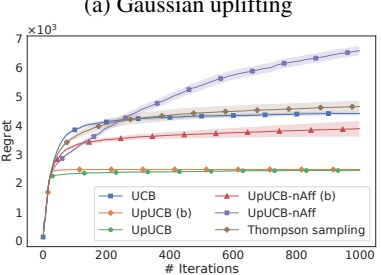

(b) Bernoulli uplifting

**Figure 3:** Experimental results on a synthetic and real-world dataset. All the curves are averaged over 100 runs and the shaded areas represent the standard errors.

tion. This dataset is designed for uplift modeling, and has outcomes for both treated and untreated individuals. Thus it is suitable for our simulations. To build the model, we sample $10^5$ examples from the dataset, and use K-means to partition these samples into 20 clusters of various sizes. The $10^5$ examples are taken as our variables. We consider 20 actions that correspond to treating individuals of each cluster, and construct independent Bernoulli payoffs using the visit rates of the treated/untreated individuals of the clusters following a procedure detailed in Appendix G.2. Here, $L = 12654$ and $\Delta$ is around 30.

**Results.** Fig. 3 confirms that we can effectively achieve much smaller regret by restricting our attention to the uplift. Moreover, when the sets of affected are known, the loss of not knowing the baseline payoffs seems to be minimal. On the other hand, not knowing the affected variables has a more severe effect in the second experiment. In fact, the design of UPUCB-nAff and UPUCB-nAff (b) heavily rely on the additive structure of the uplifting index, and can thus hardly benefit from the payoff independence which allow the other four algorithms to achieve smaller regret in this case.

This paper studies multi-armed bandit problems where the rewards are sums of observable variables. When each action only affects a limited number of these variables, much smaller regret can be achieved, and we developed algorithms with such guarantees under different forms of knowledge that the learner possesses.

While we study here a UCB-style algorithm, we believe that understanding how similar improvement can be achieved by other types of algorithms such as Thompson sampling and information directed sampling is an important question. Moreover, further extending our work to cope with non-stationary or even adversarial bandits is another promising direction to pursue. As for the former, a direct combination with existing techniques [14] can readily make our algorithms bypass the stationarity assumption that we make throughout the paper.

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
