# OpenReview forum: "Uplifting Bandits"
_NeurIPS.cc/2022/Conference — NeurIPS 2022 Accept_

### Official Review · Reviewer_b8e3 · 2022-06-23

**Rating:** 4
**Confidence:** 4
**Soundness:** 3 good
**Presentation:** 2 fair
**Contribution:** 2 fair

**Summary:**

The paper tackles the problem of detecting whether or not a strategy provides an "uplift" compared to a default one. As example applications, we have e-commerce, advertising and recommendation systems etc. where a strategy consists of a desired change from the status quo (a baseline) of a product affecting a subset of users. The goal is to determine if there is an uplift in the reward of the new strategy relative to the baseline, defined in this example as the sum of observations over all users. The setting is studied when the baseline and number of users are each known or unknown.
The paper proposes 2 algorithms (one for when the baseline reward is known and one for when the "users" affected by a strategy are unknown) accompanied by instance specific regret upper bounds. The relevance of the algorithms is demonstrated in a set of experiments on artificial data and on the Criteo Uplift dataset.

**Questions:**

When the affected arms is known, if we just use a regular UCB-algorithm that only considers the affected arms (since we know the rewards of all others will be the same), will this not result in the same kind of regret scaling as that proposed here for this setting?

How does the mechanism in UpUCB work from the perspective of the regret analysis? I would like to see this explained in the main body of the paper in order to consider raising my score for soundness. Similarly, I would have liked to see regret analysis proof architectures in the main paper. I feel the results at the moment lean too much on material deferred to the supplementary material.

Can the setting generalise to when the set affected variables change over time? For instance, if some variables are affected for one play, but others on the next etc. This would make the case where the affected variables are known more interesting and relevant in practice, in my opinion. Are there settings where we do not know the affected arms, but they are constant over time?

Is there a deeper reason why the baselines do not only consider rewards collected from affected variables (when the affected variables are known)?

**Limitations:**

I do not believe the paper has negative ethical implications.
I do believe the authors did not adequately discuss the limitations of their work.

**Strengths And Weaknesses:**

Disclaimer: I have not read the Appendix.

Strengths:
- The problem of detecting "uplift" is relevant in practice
- The paper tackles 3 variation of this setting where different information is given
- Interesting approach to examine differences of UCBs among CTRs in Algorithm 1 (UpUCB)
- The setting with unknown affected variables is the most significant of the ones presented here for practitioners
-  Numerical experiments show expected performance gain against a common-sense baseline.

Weaknesses:
- Presentation needs to be greatly improved and expose the paper's contribution over the related work more clearly. A large amount of material is deferred to the Appendix making the paper seem to not be self-contained. For instance, the results in the proofs are not obvious and I would have expected at least sketches being presented in the main paper. It is unclear why using an index based on the difference of UCBs works and I am left with a lot of questions about the quality of the evidence presented in the main body of the paper.
- The setting where both baseline and affected variables are known feels to not provide much novelty. I think this section can be sacrificed in order to provide further details about the other settings and contribution relative to prior work.
- From what I can tell, when the affected variables are known the setting reduces to the treatments affecting all variables (since we can always ignore all variables where we know there is no impact) and the regime where $m << L$ no longer holds. I believe more attention should be given to the 3rd setting where the affected variables are not known, at the expense of the others 2 settings: both baseline and affected variables are known, and only the affected variables are known.
- When the affected variables are unknown, the assumption that the same variables are always affected becomes a bit too strong to be met in real life scenarios, which I believe hurts the significance of the results. I would have liked to see much more attention given to this setting and more explanations as to why this assumption (time invariance of affected variables) is justified. As is, I feel the setting is slightly artificial and the most relevant to practice would be: a) all affected variables are known (we know which users are in an A/B test) and b) the affected variables are unknown and which variables are affected is not constant (if we don't know which users are in a test we can't guarantee returning users will make it in the same treatment).
- The baselines in the experiments seem a bit weak given that in the setting where the affected variables are known, we can just ignore all the unaffected variables.

---

> ### Author Response · Authors · 2022-08-02
> **Replies to Reviewer b8e3 (Part 2/2: Additional Content Planned Added to the Paper)**
>
> Below are our planned additions to the main body of the paper. This is feasible because if accepted, we would get one extra content page. We hope this sufficiently address your concerns on the presentation of the paper.
>
> **Idea of Proof for Theorem 1:**
>
> 1. With concentration of measure, we show that with probability $1-\delta$, it holds $|\hat{\mu}_t^a(i)-\mu^a(i)|\le c_t^a$ and $|\hat{\mu}_t^0(i)-\mu^0(i)|\le c_t^0(i)$ for all relevant estimates. It is thus sufficient to show that inequality (7) holds when these inequalities are satisfied.
> 2. A suboptimal action $a$ can only be taken if its uplifting index is larger than that of $a^{\star}$, i.e., if $\sum_{i\in\mathcal{V}^a}(U_t^a(i)-U_t^0(i))\ge\sum_{i\in\mathcal{V}^{a^{\star}}}(U_t^{a^{\star}}(i)-U_t^0(i))$. Rearranging, we get $$
>     \sum_{i\in\mathcal{V}^a}U_t^a(i)
>     +\sum_{i\in\mathcal{V}^{a^{\star}}\setminus\mathcal{V}^a}U_t^0(i)
>     \ge
>     \sum_{i\in\mathcal{V}^{a^{\star}}}U_t^{a^{\star}}(i)
>     +\sum_{i\in\mathcal{V}^a\setminus\mathcal{V}^{a^{\star}}}U_t^0(i).
>     $$ Using the inequalities mentioned in the previous point and $\Delta^a = r_{\text{up}}^{a^{\star}} - r_{\text{up}}^a$, bounding the two sides of the above inequality, we deduce $\Delta^a \le 2L^a c^a_{t-1}    +\sum_{i\in\mathcal{V}^{a^{\star}}\setminus\mathcal{V}^a}2c_{t-1}^0(i)$.
> 3. Note that for all $i\in\mathcal{V}^{a^{\star}}\setminus\mathcal{V}^a$, whenever action $a$ is taken, the count of $N_s^0(i)$ also increases by $1$. We have thus $N_{t-1}^0(i)\ge N_{t-1}^a$ and accordingly $c_{t-1}^0(i)\le c_{t-1}^a$. We then get $\Delta^a\le2(L^a+L^{a^{\star}})c_{t-1}^a$. This allows us to bound the number of times that action $a$ is played and conclude.
>
> **Idea of Proof for Theorem 2 - for Algorithm UpUCB-nAff (bl):**
>
> 1. With concentration of measure, with probability $1-\delta$ it holds $\mu^a(i)\in\mathcal{C_t}^a(i)$. Then, as explained in the text, $\widehat{\mathcal{V_t^a}}\subseteq \mathcal{V^a}$ and $\tau_t^a$ is an upper bound on $r_{\text{up}}^a$; especially $\tau_t^{a^{\star}}\ge r_{\text{up}}^{a^{\star}}$.
> 2. For $i\notin\widehat{\mathcal{V_t^a}}$, by definition $\mu^0(i)\in\mathcal{C_t^a}(i)$, from which we deduce $0\le\rho_t^a(i)\le2c_{t-1}^a$. With $\widehat{\mathcal{V_t^a}}\subseteq \mathcal{V}^a$ and $\mathcal{L_t}^a\subseteq\mathcal{V}\setminus\widehat{\mathcal{V_t^a}}$ we can then write $$
>     \tau_t^a
>     = \sum_{i\in\widehat{\mathcal{V_t}}^a\cup\mathcal{L_t^a}}\rho_t^a(i)
>     \le \sum_{i\in\mathcal{V}^a}\rho_t^a(i) + \sum_{i\in\mathcal{L_t^a}}\rho_t^a(i)
>     \le r_{\text{up}}^a+4Lc_{t-1}^a.
>     $$ We have also used $\text{card}(\mathcal{V}^a)\le L$, $\text{card}(\mathcal{L_t^a})\le L$, and $\rho_t^a(i)\le\mu^a(i)+2c_{t-1}^a$.
> 3. Whenever a suboptimal action $a$ is taken, we have $\tau_t^a\ge\tau_t^{a^{\star}}$. Then $\Delta^{a}\le 4Lc_{t-1}^a$ combining the two previous points. Subsequently, we bound the number of times that each suboptimal action $a$ is played to conclude.
>
> **Additional Related Work:**
>
> The additive structure of the reward and observability of individual payoffs suggest some similarity between our model and the semi-bandit setting (Chen et al, ICML 2013, Kveton et al, AISTATS 2015, Perrault et al., NeurIPS 2020) of combinatorial bandits (Cesa-Bianchi and Lugosi, JCSS 2012). There, the learner selects at each round a subset of
> ground items and the reward is often defined as the sum of the payoffs of the selected items. However, this seeming similarity comes with an important conceptual difference: the set of items, or variables, for which we observe the outcomes are selected in semi-bandits, while for us they are fixed and inherent to the reward generation mechanism. As an example, in Application (c) (Line 36), combinatorial bandit models optimize the number of movies that are watched among the recommended ones, while we also consider non-recommended movies. In fact, as argued in (Wang et al, RecSys 2020), a recommendation is only effective if the user actually watches the movie and would not watch it without the recommendation. We refer the readers to Appendix B for more technical details and also a comparison with the causal bandit model.

---

> > ### Comment · Reviewer_b8e3 · 2022-08-04
> > **Thanks**
> >
> > Thank you, this helps and should be included in the final version.

---

> ### Author Response · Authors · 2022-08-02
> **Replies to Reviewer b8e3 (Part 1/2: Responses to Questions)**
>
> Dear Reviewer,
>
> Thank you very much for your detailed comments and constructive feedback. To make our paper more self-contained, we will add two proof sketches and more discussion on related work to the revised version. Below, we first address the questions raised in the review, before presenting the additional content that we are going to add to our paper. Please let us know if you have any additional questions or concerns.
>
> **Can we have $L\ll m$ even if the affected variables are known?**
>
> Yes this can happen. It is true that when the affected variables are known, we can restrict our attention to the variables that are affected as the unaffected ones contribute equally to all actions. This would alter the definition of the set of variables $\mathcal{V}$ and the number of variables $m$, leading to  $m\le KL$. We have actually used this trick in the proof of Theorem 1. However, it is still possible to have $L \ll m$ provided that $K$ is large enough. Therefore, running UCB on the rewards collected from the new set of variables **will not be sufficient** (see the last reply concerning the experiments for evidences).
>
> **Regret analysis of UpUCB:**
>
> As in the analysis of standard UCB, we consider the high-probability event that each estimate falls into their corresponding confidence interval and bound the number of times that any suboptimal action is taken on this event. This is achieved using the fact that the uplifting index of the taken action is always the largest, and especially larger than the uplifting index of an optimal action. We then provide either upper bound or lower bound on the individual UCBs $U_t^a(i)$ and $U_t^0(i)$ to establish an inequality between the suboptimality gap of a suboptimal action $a$ and the sum of the widths of some confidence intervals. Thanks to the design of the uplifting indices, we end up with the widths $c_t^0(i)$ for $i\notin\mathcal{V}^a$. We show that for such $i$ it holds $c_t^0(i)\le c_t^a$ and finally establish an inequality between $\Delta^a$ and $c_t^a$. The reminder of the proof follows that of standard UCB.
>
> We will add proof sketch of Theorem 1, as outlined in our reply below.
>
> **Is it possible to generalize to the case where the affected variables change over time?**
>
> Yes, it is possible. To cope with non-stationarity in stochastic bandits, the simplest way is to either compute statistics with a sliding window (sliding-window UCB) or to compute weighted statistics (discounted UCB) -- see e.g., [Garivier and Moulines, 2008](https://arxiv.org/abs/0805.3415). It is straightforward to incorporate our methods into these approaches as only the confidence interval definitions would change. Therefore, while our work assumes stationarity, our methodology also serves as the fundamental building block for more complex methods that work in non-stationary settings. We will add a remark in our paper to clarify this point.
>
>
> **Experiments with baselines that only uses the rewards collected from affected variables:**
>
> We have chosen to run UCBs and Thompson Sampling with the reward summed over all variables for consistency with the main presentation.
>
> However, you are absolutely right that it would be more efficient to consider only affected variables. In our Bernoulli Uplifting Bandit experiment, all the variables are actually affected by one action. Therefore, there do not exist unaffected variables, but $L$ is still an order smaller than $m$, and we observe the benefit of our methods over vanilla UCB and TS, which effectively only use the rewards collected from affected variables.
>
> As for the Gaussian Uplifting Bandit Experiment, 68 out of the 100 variables are affected (by at least one action). We rerun experiments for UCB and TS with rewards collected from these 68 variables. You can find these new results in Appendix H.3 of the revised supplementary. We observe that the proposed methods still achieve much smaller regret, confirming that simply ignoring the unaffected variables may not be sufficient when there are many actions.

---

> > ### Comment · Reviewer_b8e3 · 2022-08-04
> > **Further clarifications**
> >
> > Thank you for your detailed answer!
> >
> > Unfortunately I believe there is a typo in my review (regime mentioned is $m << L$ instead of $L << m$) which raises a bit of confusion when trying to reconcile your answer. I hope it is possible to clarify the following point once more, just to make sure my understanding is accurate:
> >
> > To clarify my understanding: $L$ is the (upper bound on the) number of variables affected by one arm, $m$ total number of variables, $K$ total number of arms (each arm can have a different uplift on each variable it affects). In the first setting, for each arm, we know (and observe the pay-offs of) the set variables it affects (whose rewards it changes) and the baseline. My main concern is surrounding the possibility that for a given arm we can only consider the variables affected by this specific arm when gathering reward statistics (so the fact that one arm affects all $m$ variables should not matter for all other arms), and not all the $m$ variables that are affected by at least one arm.
> >
> > In my understanding, we observe the components of the reward (the payoff vector) that correspond to these variables - so we can compute the reward only using the components corresponding to the affected variables (hence $L \approx m$ for every arm). This is also what I would have liked to see in the baseline algorithms. If my understanding is correct, we don't need to know the individual payoff per variable to determine the overall pay-off of an arm - we just need to find the uplift of the sum over only the affected variables of each arm, is this correct? Would this translate into a classical bandit setting with $K$ arms? How is the structure of the problem (known affected variables) helpful for reducing regret in this scenario relative to this classical bandit equivalent?
> >
> > I appreciate any perspective you can provide to clarify this distinction.

---

> > > ### Author Response · Authors · 2022-08-04
> > > **What you propose here is UpUCB (bl), and you are right we only need to observe the sum over the variables each arm affects**
> > >
> > > Dear Reviewer,
> > >
> > > Thank you very much for your prompt reply and detailed feedback. We greatly appreciate it.
> > >
> > > As far as we understand, what you are proposing here is exactly equivalent to the UpUCB (bl) algorithm described in Section 3. You are right in saying that for each action $a$, we only need to observe the reward summed over $\mathcal{V^a}$, the variables affected by this action. Then, we apply classic bandit algorithms to estimate the uplift $r_{\text{up}}^a=\sum_{i\in\mathcal{V^a}}(y^a(i)-\mu^0(i))$. There is no need to compute each single $U^a_t(i)$, but mathematically, this is the same algorithm as UpUCB (bl).
> > >
> > > The important observation is thus that, while the original reward is defined as the sum over all the $m$ variables (such as the total revenue summed over all the customers, including those that are not affected by action $a$, so generally $L^a\ll m$), we achieve much smaller regret by focusing on the variables that each arm affects, as what you are suggesting here. The main contribution of our work is then to go beyond this simple $K$-armed bandit algorithm by designing methods for the cases where the baseline or the sets of affected variables (or both) are unknown.
> > >
> > > We will clarify this point in our revision. Please let us know if you have further questions. Thank you again for engaging in the discussion.

---

### Official Review · Reviewer_wS6g · 2022-06-29

**Rating:** 6
**Confidence:** 4
**Soundness:** 2 fair
**Presentation:** 3 good
**Contribution:** 2 fair

**Summary:**

The paper studies a MAB problem where the reward has an additive structure. Each arm affects a subset of terms of the overall reward, and the goal is to maximize reward while not knowing which subsets are affected (e.g, their reward distributions changing) by which arms.


**Questions:**

See above.

**Limitations:**

Yes.

**Strengths And Weaknesses:**

From one view, this problem can be thought of as a combinatorial bandit problem where you have n sets to choose from and each set leads to a different payoff. What difference does a baseline different from 0 make in the analysis and development of the online algorithm? If the baseline is 0, isn't this the same as the set optimization problem? Given these two points of view, the paper does not make it clear why the general combinatorial bandit strategies do not apply. In fact some of them explicitly make use of the objective function structure (linear, Lipschitz etc) and derive regret bounds. (The individual payoffs of elements in the set are assumed to be observable in the paper whereas its not needed in the general case.) It is hard to make out from the paper what the significant differences and disadvantages are of applying these ideas? Or is it that the structure proposed in the paper does not map to a clean offline combinatorial problem at all?

Can the authors clarify: "In contrast to standard UCB methods, these differences are not optimistic". What does this mean (I am likely missing a nuance here)?

Since uplift modeling is a popular approach in causal inference (and applied to all the usecases mentioned in the paper), this causes wrong expectations to be set and misleads the reader (even though the authors mention this and cite 11,16).

---

> ### Author Response · Authors · 2022-08-02
> **Replies to Reviewer wS6g**
>
> Dear Reviewer,
>
> Thank you very much for your insightful questions. We address them below. Please let us know if you have any additional questions or concerns.
>
> **Relation to Combinatorial Bandits:**
>
> To begin, there is an important conceptual difference between our model and that of combinatorial bandits. In the latter, we select explicitly the set of items that represents the action and the reward is defined as the sum over the individual payoffs of the *selected items*, while in the former (ours), each action only affects a small number of variables (note that this set is not necessarily known, see below), and the reward is summed over *all the $m$ variables*. These two models are introduced with different motivations, and also have different technical implications.
>
> In order to reduce our model to a combinatorial bandit one, each action thus needs to contain exactly $m=\text{card}(\mathcal{V})$ items. As the actions affect the variables differently, the ground set of items $E$ has the following members:
>
> 1. For each variable $i \in \mathcal{V}$, we have an item $e_{0, i}$ with mean $\mu^0(i)$. This item models the reward of variable $i$ when it is unaffected by the taken action.
> 2. For each action $a\in\mathcal{A}$ and $i \in \mathcal{V}^a$, we have an item $e_{a, i}$ with mean $\mu^a(i)$. This item models the reward of variable $i$ when affected by action $a$.
>
> Then, when action $a$ is taken, we observe the stochastic reward of $e_{a, i}$ for $i \in \mathcal{V}^a$, and that of $e_{0, i}$ for $i \notin \mathcal{V}^a$. The total reward is the sum of the observed individual rewards.
>
> We would like to make three comments on this reduction.
>
> - First, when the baseline is known, we can ignore the items $e_{0,i}$ but this requires to shift the rewards of the items when the baseline is non-zero as explained in our text.
>
> - Second, when the baseline is unknown, we need to estimate the rewards of all the items. Through a transformation, UpUCB can be reformulated as CombUCB on the reduced model. However, the general analysis for semi-bandits (as in Kveton et al. AISTATS 2015) yields a regret of $O(m^2+mKL)$ in this case as there are $m+KL$ items (assume here each action affects exactly $L$ variables) and each action contains $m$ items, while our analysis gives a regret of $O(KL^2)$. This leads to great improvement when $L\ll m$, and this is made possible by exploiting the particular structure of the problem.
>
> - Finally, one of the most important contribution of our work is consider the situations where the sets of affected variables are unknown. In the transformed model, this means that we do not know the composition of each action. We are not aware of any works that address this situation in the combinatorial bandit liteurature.
>
> A version of the above discussion already appears in Appendix B. We will add a paragraph in introduction to further clarify this point.
>
> **UpUCB is not Optimistic:**
>
> UCB algorithms are said to be optimistic because with high probability the UCB indices are upper bounds on the actual expected rewards. Then, when the learner is choosing their action, they are *optimistic* about the rewards they will receive (they overestimate these rewards). In contrast, the uplifting indices that we construct in UpUCB are not high-probability upper bounds on the actual uplifts, so the algorithm is not optimistic in the sense of UCB. We will clarify this point.
>
> **On the Term Uplift:**
>
> While the taken approach may be different, the fact that each action causes an uplift with respect to a certain baseline is crucial to both our model and algorithmic design. We have made an effort to be explicit in our model and definitions in Section 2. We hope this should cause no confusion.

---

### Official Review · Reviewer_xacy · 2022-07-16

**Rating:** 6
**Confidence:** 3
**Soundness:** 3 good
**Presentation:** 3 good
**Contribution:** 3 good

**Summary:**

This paper introduces the framework of uplifting bandits - wherein an action affects a subset of intermediate variables and the rewards are the sum of the random variables. The uplift of an action is then computed by comparing it against the baseline where no variable is affected. Variants of the UCB algorithm are proposed that use varying amounts of knowledge about the structure of the problem and result in improvement in regret.

**Questions:**

To my understanding, if the covariance matrices in the “Gaussian Uplifting Bandit” experiment were to be changed to be (significantly) different for each action, the structure agnostic algorithms (UCB and TS) would perform relative better than the UCB variants. Do the authors have ane experiments about the same, or a comment on the sensitivity of the algorithms this variance in the noise covariance matrix?

**Limitations:**

Yes, the limitations have been addressed by the authors. The potential negative societal impact of these algorithms in the applications mentioned has not been discussed.

**Strengths And Weaknesses:**

Strengths:
- Introduces a bandit framework that better models some applications mentioned in the paper
- The paper is clearly written and the illustrations make the dense material easy to follow
Weaknesses:
- The algorithms proposed rely on a significant amount of knowledge of the structure of the problem that is often not realisable in real settings. Figure 3(b) shows an example where knowledge and lack thereof make a large difference in performance compared to methods completely agnostic of the structure of the problem

L122: achieve (typo)

---

> ### Author Response · Authors · 2022-08-02
> **Replies to Reviewer xacy**
>
> Dear Reviewer,
>
> Thank you very much for your thoughtful feedback and positive evaluation. We address your concerns below. Please let you know if you have any further questions.
>
> **On use of knowledge about the structure:**
>
> It is true that some knowledge about the problem is required for achieving improved regret, and we have actually shown the necessity of such knowledge through our lower bound (Section 4). Clearly, the less we know about the problem, the more difficult it is to improve the efficiency of the algorithm. Nonetheless, we believe that our regret bounds can provide guidance on when such improvement is plausible.
>
> As for Figure 3(b), with the independence of the noises, the regrets of UCB and TS scale as $O(\sqrt{m})$. At the same time, the regret of UCB-nAff necessarily scales as $O(L)$ due to the heavy use of the additive structure of the uplifting indices in the algorithm. Therefore, with $\sqrt{m}=10^{2.5}<L=12654$ in the experiment, it is expected that UCB and TS would perform better than UpUCB-nAff. This is an example where preliminary insights can be used to decide whether our algorithm would bring improvement or not. Note that when the noises are correlated, our schemes clearly outperform UCB and TS.
>
> **On use of different covariance matrices:**
>
> Regarding the Guassian Uplifting Bandit experiment, we do not think our methods would lose their advantage when the covariance matrices of the actions were significantly different, as focusing on fewer variables always helps improving the statistical efficiency. To confirm our hypothesis, we construct a Gaussian Bandit instance in which the actions have very different covariance matrices, resulting in total variances that range from 16 to 80. We then run all the considered algorithms on this problem. Please refer to the results in Appendix H.4 of the revised supplementary. Here we see again that the methods that exploit the structure achieve smaller regret, with an improvement comparable to the initial experiment.

---

> > ### Comment · Reviewer_xacy · 2022-08-08
> > **Thanks for your responses**
> >
> > Thank you, this helps answer the questions above. I will maintain the score.

---

### Official Review · Reviewer_H2gA · 2022-07-18

**Rating:** 6
**Confidence:** 4
**Soundness:** 4 excellent
**Presentation:** 4 excellent
**Contribution:** 3 good

**Summary:**

This paper studies a variant of multi-armed bandits, where each action can change the distribution of some random variables (which otherwise follow baseline distributions), and the observed reward is a sum of these random variables. The learner's goal is to find the best action while keeping the cumulative regret as low as possible.

To do this, the authors use the ideas from the uplifting modeling (modeling the incremental effect of an action on an individual variable) and propose upper confidence bound (UCB)-based algorithms that estimate uplifts over a baseline distribution. They prove that all algorithms have sub-linear regret upper bounds and empirically validate the performance of algorithms.

**Questions:**

Please address the weakness raised in **Strengths And Weaknesses**. I can change my score based on the authors' responses.

**Limitations:**

The lower bound given in Section 4 (already similar results exist for combinatorial semi-bandits) and contextual extensions consider in Section 7 are not significant results. Since this work is a theoretical paper, I do not find any direct negative societal impact.

**Strengths And Weaknesses:**

#### **Strengths of paper:**
1. The problem setting is well motivated. The authors consider different settings (i.e., known or unknown affected variables and baseline payoffs) and develop new ideas (either exploiting underlying problem structure or adding some mild assumptions) to design algorithms for these settings.

2. The authors propose upper confidence bound (UCB) based algorithms named UPUCB (bl), UPUCB, UPUCB-nAff(bl), and UPUCB-nAff for four different settings. These algorithms exploit underlying problem structure (and additional assumptions for the most challenging setting, i.e., unknown affected variables and baseline payoffs) and show that these algorithms have sub-linear regret.

3. The authors have empirically validated the different performance aspects of the proposed algorithms.


#### **Weakness of paper:**
1. The setting when the affected variable and baseline payoffs are known can be modeled as a special case of the combinatorial bandit problem, which is the simplest form of the problem (also discussed in the supplementary material). I notice that the authors have missed the most recent work in this direction by Perrault et al., 2020 (Paper: Statistical Efficiency of Thompson Sampling for Combinatorial Semi-Bandits, NeurIPS 2020).

2. Since the lower bound for the simplest case can be adapted from the lower bound given in Perrault et al., 2020. Hence, it is unclear how the lower bound presented in the paper is different.

3. The contextual extension discussed in Section 7 is a simple extension when the affected variables and baseline payoffs are known. Even for other settings, after having the underlying ideas for non-contextual cases, I am not sure what are the difficulties to extend for contextual setting (as estimating reward needs to be changed and for that exiting contextual bandit's results can be used).

---

> ### Author Response · Authors · 2022-08-02
> **Replies to Reviewer H2gA**
>
> Dear Reviewer,
>
> Thank you very much for your thoughtful input and detailed comments. We address below the weaknesses raised in the review.
>
> **Relation to Semi-Bandits:**
>
> We have already added the missing reference Perrault et al., NeurIPS 2020 in the revised version. Thank you for pointing us to this nice paper.
>
> For completeness, we provide below more clarification on how our model relates to and differs from semi-bandits (as also discussed in Appendix B).
>
> To begin, there is an important conceptual difference between our model and that of semi-bandits. In the latter, we select explicitly the set of items that represents the action and the reward is defined as the sum over the individual payoffs of the *selected items*, while in the former (ours), each action only affects a small number of variables (note that this set is not necessarily known, see below), and the reward is summed over *all the $m$ variables*.
>
> In order to reduce our model to a semi-bandit one, each action thus needs to contain exactly $m=\text{card}(\mathcal{V})$ items. As the actions affect the variables differently, the ground set of items $E$ has the following members:
>
> 1. For each variable $i \in \mathcal{V}$, we have an item $e_{0, i}$ with mean $\mu^0(i)$. This item models the reward of variable $i$ when it is unaffected by the taken action.
> 2. For each action $a\in\mathcal{A}$ and $i \in \mathcal{V}^a$, we have an item $e_{a, i}$ with mean $\mu^a(i)$. This item models the reward of variable $i$ when affected by action $a$.
>
> Then, when action $a$ is taken, we observe the stochastic reward of $e_{a, i}$ for $i \in \mathcal{V}^a$, and that of $e_{0, i}$ for $i \notin \mathcal{V}^a$. The total reward is the sum of the observed individual rewards.
>
> We would like to make three comments on this reduction.
>
> - First, when the baseline is known, we can ignore the items $e_{0,i}$ but this requires to shift the rewards of the items when the baseline is non-zero as explained in our text.
>
> - Second, when the baseline is unknown, we need to estimate the rewards of all the items. Through a transformation, UpUCB can be reformulated as CombUCB on the reduced model. However, the general analysis for semi-bandits (as in Kveton et al. AISTATS 2015) yields a regret of $O(m^2+mKL)$ in this case as there are $m+KL$ items (assume here each action affects exactly $L$ variables) and each action contains $m$ items, while our analysis gives a regret of $O(KL^2)$. This leads to great improvement when $L\ll m$, and this is made possible by exploiting the particular structure of the problem.
>
> - Finally, one of the most important contribution of our work is consider the situations where the sets of affected variables are unknown. In the reduced model, this means that we do not know the composition of each action. We are not aware of any works that address this situation in the semi-bandit literature.
>
>
> **Lower Bounds:**
>
> Since our model is only a special case of the general semi-bandit model, a lower bound for semi-bandits does not "automatically" imply a lower bound for our model. However, it turns out that the reduction used in Kveton et al., AISTATS 2015 can effectively be adapted to prove the statement in Part (a) of Proposition 2 (in Appendix D.1 after Proposition 5 we have briefly mentioned the possibility of applying such reduction). On the other hand, the Parts (b) and (c) of Proposition 2 are very different from existing lower bounds for semi-bandits, so no such adaptation seems possible.
>
> **Contextual Extensions:**
>
> We totally agree that as long as a UCB can be constructed for the quantity $f^a(x_t(i))$, our algorithms can be directly extended to the contextual setup considered in Section 7. We believe this is an important advantage of our method as many bandit algorithms do not easily generalize beyond the vanilla MAB setup. The goal of Section 7 was thus convey this idea through simple examples. While the algorithms are easy to design, their regret would depend on how efficiently we can estimate the parameters using the obtained information. This would thus require a case-by-case analysis that varies across the choice of $f^a$ that is beyond the scope of this work.
>
>
> We hope that the above comments sufficiently address your concerns. If you have any additional questions, please let us know.

---

> > ### Comment · Reviewer_H2gA · 2022-08-09
> > **Rebuttal Reply**
> >
> > Thank you for your detailed rebuttal. It clarified and addressed all my questions. I keep my score as it is.

---

### Author Response · Authors · 2022-08-05
**Ready to discuss the paper and engage in a discussion**

Dear reviewers,

We wanted to thank you for insightful and detailed reviews. This allowed us to put together what we believe is a good rebuttal and we submitted it 3 days ago. If you have any additional questions or concerns, we would love to hear from you and engage in a discussion. We realize that the paper covers a lot of content and your feedback so far helped us to rethink how to present it more clearly.

Sincerely,

The authors

---

### Meta-Review · Area_Chair_Njru · 2022-09-02

**Recommendation:** Accept
**Confidence:** Certain

**Metareview:**

The paper's main contribution is, in my opinion, conceptual. It introduces a new multi-armed bandit problem with arms ('interventions') that affect a sparse set of downstream variables, a 'baseline effect' and an additive reward ('uplift') structure. The motivation for this is drawn from uplift modeling and its associated causal inference from marketing and e-commerce domains.

While this could, on one hand, be cast aside as 'yet another' combinatorial-type bandit problem, I believe that it brings to the table an interesting and thought-provoking information and decision structure to reason about, setting it apart from typical variations of bandit problems. Though one of the reviewers questions its scope as being rather narrow, I believe it is worth taking a risk ('pulling an exploratory arm'!) and see this paper published in the hope that it spurs other fruitful work in this interface of marketing and ML.

I note also that the paper is written in a clear and lucid fashion, and urge the author(s) to suitable incorporate their suggested additions and clarifications arising from the engagement with the reviewers into the final version.

**Award:**

No

---

### Decision · Program_Chairs · 2022-09-14

Accept